# White-tailed deer population declines in a high-prevalence chronic wasting disease region of Arkansas, USA

Heather E. Gaya[1]*, Marcelo H. Jorge[1], Lisa A. Jorge[1], Mark G. Ruder[2], Gino J. D'Angelo[1], Richard B. Chandler[1], Michael J. Chamberlain[1]

**1** Warnell School of Forestry and Natural Resources, University of Georgia, Athens, Georgia, United States of America, **2** Southeastern Cooperative Wildlife Disease Study, University of Georgia, Athens, Georgia, United States of America

* heather.gaya@uga.edu

## Abstract

Chronic wasting disease (CWD) is a fatal transmissible spongiform encephalopathy affecting cervids worldwide. CWD was first detected in Arkansas in 2015 and as of August 2025 has been detected in 24 counties across the state. Within the Tier 1 CWD management zone of northern Arkansas, average apparent CWD prevalence exceeded 25% at the onset of our study in 2021. We tested the hypothesis that high prevalence of CWD negatively affects white-tailed deer population viability. We collected data from 243 camera traps and deployed GPS-collars on 131 adult deer to monitor population dynamics. Using spatial mark-resight models, we estimated density of adult deer from 2021 to 2024 at three sites across a presumed CWD gradient to assess the impacts of high CWD prevalence on deer abundance. Deer densities declined at all three study sites, at an average 17% (95% CI: 8% − 24%) decline per year. Male densities declined by an average 23% (95% CI: 5% − 31%) per year compared to 15% (95% CI: 2% − 23%) yearly declines for females. These findings suggest that CWD can negatively impact deer populations through direct reductions in density, but additional research is needed to determine if additional factors contributed to these declines. Furthermore, our findings suggest the populations we studied are not sustainable under current harvest regulations.

## Introduction

Chronic wasting disease (CWD) is a fatal, contagious prion disease of white-tailed deer (*Odocoileus virginianus*), mule deer (*Odocoileus hemionus*), elk (*Cervus elaphus*), moose (*Alces alces*), and some other members of the cervid family. The disease is caused by the accumulation of a misfolded protein (PrPSc) in lymphoid and brain tissues after a prolonged and variable incubation period (typically 16–24 months), ultimately leading to neurodegeneration and host death [1,2]. The first accounts of CWD occurred

**Data availability statement:** All code relevant code and data are publicly available on Zenodo: https://doi.org/10.5281/zenodo.17594064.

**Funding:** Funding for this project was provided by the United States Forest Service, CWD Alliance, Cabela Family Foundation, Boone and Crockett Club, the Wildlife Restoration Program through the U.S. Fish and Wildlife Service, and the Arkansas Game and Fish Commission (AR-W-F20AF00265). The funders had no role in study design, data collection and analysis, decision to publish, or preparation of the manuscript.

**Competing interests:** The authors have declared that no competing interests exist.

during the late 1960s in captive mule deer from Colorado and Wyoming but the clinical syndrome was not confirmed as a transmissible spongiform encephalopathy until 1978 [3]. CWD has since been detected in free-ranging cervid populations in 36 states and four Canadian provinces (U.S. Geological Survey 2025). CWD has been linked to population declines in both mule deer and white-tailed deer [4–6], raising concerns about deer population viability in CWD endemic areas. However, few studies have directly investigated population-level impacts of CWD on free ranging cervid populations.

The first detection of CWD in a free-ranging white-tailed deer in Arkansas was a clinically affected 2.5-year-old doe from Newton County in 2016 [7]. Subsequent surveillance via sharpshooting conducted immediately after the first detection suggested apparent prevalence was 23%, with 25% prevalence for deer less than one year of age [8]. As of 2025, CWD has been detected in 23 additional counties across the state, with a 28% average apparent prevalence within the 5 counties surrounding the initial detection location [7]. Given the widespread high prevalence in this area, understanding how CWD impacts population abundance and growth rates will be critical for establishing sustainable harvest regulations.

Abundance estimation is a critical component of successful population management, particularly when harvest limits are dependent on current population size [9,10]. State wildlife agencies routinely collect harvest data at the county or management unit scale to monitor white-tailed deer abundance indices, but costs associated with collecting population-level data often preclude rigorous abundance estimation [11,12]. Camera traps are a cost-effective method of collecting information on large vertebrates across broad spatial scales but the resulting data can be challenging to analyze if wildlife species lack unique individual markings. When species are individually identifiable, either with tags or natural unique markings, camera data can be modeled with spatial capture-recapture (SCR) models to make inference about population density [13,14]. Recent extensions to SCR models, referred to as spatial mark-resight models (SMR), relax the requirement for individual markings, allowing for inference on abundance and density from camera data when populations contain both marked and unmarked individuals [15–17]. This technique is especially effective when a subset of the population has been marked with GPS telemetry devices [16,18]. Several studies have applied SMR models to data on male white-tailed deer and fawns [19–21], but few studies have used the methodology with female deer.

Our goal was to implement an SMR model for both male and female deer to test the hypothesis that high prevalence of CWD negatively impacts white-tailed deer population density. We tested our hypothesis using GPS data and camera trap data collected across a 4-year study period at 3 sites in northern Arkansas across a CWD gradient. We hypothesized that deer densities would decline across the study period in response to CWD. In accordance with this hypothesis, we predicted that deer densities would be negatively associated with CWD prevalence.

## Methods

We conducted research across 3 sites in northwestern Arkansas on the border of the Ozark highlands in Searcy and Newton Counties (Fig 1). The sites were chosen

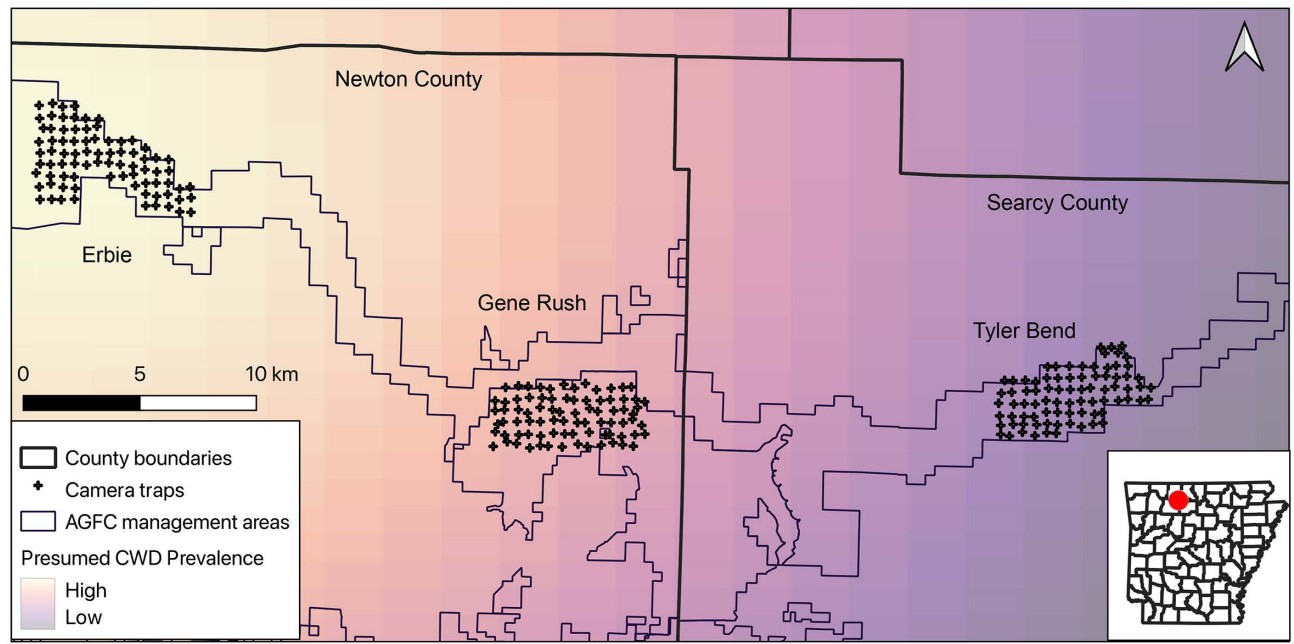

**Fig 1. Map of study area showing camera trap array at 3 sites in Northern Arkansas, United States.** Black crosses represent camera traps (N = 243) operated from November 2020 to March 2025. Light polygons represent Arkansas Game and Fish Commission (AGFC) wildlife management areas, with darker lines indicating county borders. Background color represents an assumed CWD prevalence gradient from east to west based on county-level CWD prevalence estimates reported by AGFC.

to represent an east to west gradient of declining apparent CWD prevalence based on the density of CWD detections in white-tailed deer collected by the Arkansas Game and Fish Commission (AGFC). The 3 sites included 2 National Park Service sites (Erbie and Tyler Bend) and one state Wildlife Management Area (Gene Rush Wildlife Management Area [WMA]). Vegetation at the sites was dominated by oak-hickory forests, with oak-hickory-pine forests present on steep slopes. Dominant tree species included northern red oak (*Quercus rubra*), southern red oak (*Q. falcata*), white oak (*Q. alba*), and multiple hickory species (*Carya spp.*), with the occasional shortleaf pine (*Pinus echinata)* growing on drier south and west facing slopes.

Spatial mark-resight models benefit from GPS collar data, which when combined with camera traps, provide direct information about the probability that an individual will be detected by a camera, conditional on the location of its home range. We captured white-tailed deer adults (≥1.5 years old) using rocket nets, drop nets (Wildlife Capture, Flagstaff, Arizona, USA), Clover traps (Wildlife Capture, Flagstaff, Arizona, USA), and by remote chemical immobilization with a dart gun (Pneu-Dart, Williamsport, Pennsylvania, USA) in the winters of 2021–2024. Captured deer were immobilized with intramuscular injections of butorphanol tartrate (27.3 mg/ml), azaperone tartrate (9.1 mg/ml), and medetomidine hydro-chloride (10.9 mg/ml; BAM; Zooparm, Fort Collins, Colorado, USA). Additional details on capture protocols can be found in Jorge (2024). All captured deer were given ear tags and adults were fit with GPS collars (Vectronic Aerospace, Berlin, Germany). Capture and handling protocols were approved by the University of Georgia Institutional Animal Care and Use Committee (approval A2023 05–030-Y1-A0), and Arkansas Scientific Collection Permit (081320202).

We programmed GPS collars to take fixes every six hours year-round, except during the rut and peak fawning periods. We switched collars to a 2-hour fix rate from October 15th – December 15th for adult males and from May 1st – July 31st for adult females to allow for finer-scale assessments of movements. All collars were set to remotely drop after two years if no mortality was detected.

We deployed 81 camera traps (Hyperfire 2, Reconyx, Holmen, WI) at each of the three study sites for a total of 243 cameras. We placed cameras in a systematic design within each site to maximize detection of unique individuals. We separated trail cameras by approximately 500m and placed them approximately 40 cm high on trees along game trails, hiking trails, roads, and field edges to maximize deer detections. Cameras recorded data throughout the year from November 2020 to March 2025. We checked cameras monthly to download data, replace batteries, and clear vegetation. We used MegaDetector [22], a free and open-source object detection model, to determine if there was an object in the photo, and then used MegaClassifier [22] to identify species. We further developed a custom object detection algorithm using the Ultralytics package in Python [23] to classify photos of deer by sex and age (fawn or adult) and flag photos of deer with ear tags or collars. We then manually checked detections and classifications for mistakes and uploaded images of target species to Camelot Project photo management software [24].

Previous research suggests that deer densities in the Southeast, US are negatively associated with large areas of cultivated crops, pastures and grasslands [25]. We split the study area into 2-km by 2-km pixels to reflect the average summer home range size of white-tailed deer in the area [26]. In each pixel, we calculated the percent pasture, crops, and open fields in each year using the National Land Cover Database [27]. Given the high CWD sample prevalence in Newton County relative to Searcy County (one county east) or Stone County (two counties east) [28], we assumed a west to east gradient in CWD prevalence across the study area. To account for the presumed CWD gradient, we assigned each pixel an Easting value, which we scaled for analysis by subtracting the mean easting for the study area from the easting coordinate of each pixel and dividing by the standard deviation. We considered this a proxy for distance to the first known CWD detection in Arkansas, as locations of positive detections in Arkansas were not publicly available. We modified the framework for a two-stage SMR model [17] to model adult white-tailed deer density across our sites from July 2021 – July 2024 by combining camera and GPS collar data. Female deer tend to reduce home range size, isolate themselves from conspecifics, and decrease movement rates in the weeks before and after parturition [29,30]. To maximize detections of collared deer in camera photos, we chose to analyze deer densities from July 1 – July 14th of each year, representing roughly 40 days after peak fawning [31]. During this time frame, bucks, fawns and does are easily distinguishable from one another.

In a two-stage SMR model, individuals with known locations (e.g., GPS collared deer) are used to estimate the detection parameters in a standard SCR framework. The posterior distribution for each detection parameter is then used as a prior in the second stage, which uses camera data to estimate abundance, treating all individuals as unmarked. In stage 1, we modeled the expected number of detections of each collared deer as a half-normal function, where the number of detections of individual $i$ of sex $k$ at camera $j$ in year $t$ was assumed to decrease as the distance, $d_{i,j}$ between camera $j$ and the individual's activity center increased: $\lambda_{i,k,j,t} = \lambda_{0,k,j,t} * e^{-d_{i,j}/(2*\sigma^2_{k,j,t})}$. The rate of decline in detection probability with distance was controlled by the spatial scale parameter, $\sigma_{k,j,t}$. When the distance between the camera and the activity center was 0, the expected number of detections reverted to the baseline encounter rate, $\lambda_{0,k,j,t}$.

To account for differences in detection between sites and sexes across years, we modeled both the spatial scale parameter and the baseline encounter rate as log linear equations. The baseline encounter rate for each year, $\lambda_{0,k,j,t}$, was modeled as a mean with a fixed effect for both site and sex

$$\log(\lambda_{0,k,j,t}) = \beta_0^\lambda + \beta_1^\lambda [site_j] + \beta_2^\lambda * sex \qquad (1)$$

An identical equation was used to calculate $\sigma$ (Table 1). We did not differentiate between yearling and adult individuals as white-tailed deer, particularly females, cannot be reliably aged in camera photos. We used a bivariate normal to model each deer's daily GPS location relative to its activity center, $s_{i,k,,t}$, in that year: $u_{i,j,k,t} \sim Normal\left(s_{i,k,,t}, \sigma^2_{i,j,k,t}\right)$. The encounter history for each individual at each trap in each year across the 2-week period was modeled as: $n_{i,j,k,,t} \sim Binomial\left(14, 1 - e^{-\lambda_{i,j,k,t}}\right)$.

 

**Table 1. Priors used in a spatial mark resight model of white-tailed deer.**

| Parameter | Description | Prior | |
|---|---|---|---|
| $\alpha_0$ | Intercept for expected abundance | Normal(0, 1) | |
| $\alpha_1$ | Relationship between abundance and percent pasture and grassland cover | Normal(0, 1) | |
| $\alpha_2$ | Relationship between abundance and Easting (proxy for CWD gradient) | Normal(0, 1) | |
| $\alpha_{3,k}[site]$ | Sex and site specific annual trend on expected abundance | Normal(0, 1) | |
| $\beta_0^\lambda$ | Intercept for baseline detection probability | Normal(log(0.1), 1) | |
| $\beta_1^\lambda[site]$ | Effect of site on baseline detection probability | 0, | Site = Erbie |
| | | Normal(0, 1), | Otherwise |
| $\beta_2^\lambda$ | Effect of sex = male on baseline detection probability | Normal(0, 1) | |
| $\beta_0^\sigma$ | Intercept for detection scale parameter | Normal(log(350), 1) | |
| $\beta_1^\sigma[site]$ | Effect of site on detection scale parameter | 0, | Site = Erbie |
| | | Normal(0, 1), | Otherwise |
| $\beta_2^\sigma$ | Effect of sex = male on detection scale parameter | Normal(0, 1) | |

Priors used in a spatial mark resight model for deer abundance in northern Arkansas, United States from 2021 to 2024. All parameters are on the log scale.

In the second stage of our SMR model, we modeled the activity centers of each deer, $s_{i,k,t}$, in each year $t$ using an inhomogeneous Poisson point process based on the percent of pasture and open fields, $pasture_{q,t}$, in each pixel $q$, a scaled easting coordinate, $easting_q$, and a site and sex specific time trend. The expected number of deer in each time period was the sum of the expected density of deer in each pixel $q = 1,\ldots, Q$,

$$E\left(N_{k,t}\right) = \sum_{q=1}^{Q} \mu_{q,k,t} * Area \tag{2}$$

$$log\left(\mu_{q,k,t}\right) = \alpha_0 + \alpha_1 * pasture_{q,t} + \alpha_2 * easting_q + a_{3,k}\left[site_q\right] * \left(t-1\right) \tag{3}$$

where *Area* is the size of each pixel in square kilometers.

When using camera data on unmarked individuals, it is difficult to determine which photos are independent observations, since deer frequently stop in front of cameras and may be detected in several consecutive photos. To avoid the need to model the dependency between photos, we chose to analyze the data as a binary for each 24-hour period. Assuming the number of independent observations of each deer was Poisson distributed, the probability of camera $j$ detecting deer $i$ of sex $k$ at least once during occasion $t$ is: $1 - e^{-\lambda_{i,j,k,t}}$. Thus, for any given camera $j$, the probability of detecting at least one deer during occasion $t$ can be modeled as:

$$n_{j,k,t} \sim Bernoulli\left(1 - e^{-\sum_{i=1}^{N_{k,t}} \lambda_{i,j,k,t}}\right) \tag{4}$$

However, this formulation requires knowledge of $N_{k,t}$, the true abundance of the population, which we did not know *a priori*. To resolve this, we used a data augmentation framework [15,32] where we proposed a value $M$, which was an integer value much larger than the true abundance, $N_{k,t}$. We defined $z_{i,k,t}$ as a binary variable that indicated if an individual was real and present in the population at time $t$. The probability that individual $i$ was real ($z_{i,k,t} = 1$)

was noted by $\psi_{k,t} = \frac{E(N_{k,t})}{M}$. Thus, as the expected value of abundance increases, so too does the probability that an individual is real and in the population. Under data augmentation, we calculated realized abundance as:

$$N_{k,t} = \sum_{i=1}^{M} z_{i,k,t} \qquad (5)$$

$$z_{i,k,t} \sim \text{Bernoulli}\,(\psi_{k,t}) \qquad (6)$$

Yearly site-specific realized abundances were calculated as the sum of real individuals ($z_{i,k,t} = 1$) of each sex with activity centers ($s_{i,k,,t}$) located within the bounds of each study site. We further estimated yearly changes abundance at each site by dividing the total yearly abundance at each site by the abundance in the previous year.

We fit both stages of the model in NIMBLE using package *nimbleSCR* [33,34] in program R [35]. We ran each stage of the model for 75,000 iterations with a burn-in of 60,000, resulting in 15,000 posterior samples. We report parameter estimates as posterior means and 95% credible intervals.

## Results

From July 2021 to July 2024, camera traps recorded > 4 million photos, of which 2,148,250 were classified as containing deer. We further subset these photos into images of adult deer taken between July 1 and July 14 of each year, for a final sample of 46,314 photos across 243 cameras. Collared deer were detected in 1,228 photos. We used detection histories and GPS-collar locations from 131 deer (52 M, 79 F) to estimate site, sex and year specific encounter rate parameters (S1 Table). The number of GPS collared deer at each site ranged from 4 to 38 deer per year, with females representing most (66–75%) of the collared deer in each year (Table 2). Based on AGFC sampling records, CWD apparent prevalence in Newton County (where Erbie and Gene Rush sites were located) rose from 34% in 2020–2021 to 49% in 2022–2023, whereas apparent prevalence in Searcy County (Tyler Bend) increased from 11% in 2020–2021 to 20% in 2022–2023 (S1 Fig). To calculate the scaled Easting value that we used as a proxy for the CWD gradient, we used a mean value of 501979.4 and a standard deviation of 17070.23.

Across all years, deer density was highest at Tyler Bend and lowest at Erbie, inversely mirroring trends in CWD sample prevalence (Fig 2). We found a positive relationship between deer densities and distance to Arkansas's first known CWD detection, suggesting deer densities were highest on the eastern side of the study area (Fig 3). There was a negative relationship between percent pasture cover and deer densities across all sites (Table 3). Median deer densities declined from 2021 to 2024 at all three study sites, with an average 17% (95% CI: 8% − 24%) annual decline across the entire study area. At Erbie, the average annual decline was 17% (95% CI: −3% − 33%), but the credible interval included 0. At

**Table 2. Number of GPS collared adult deer at three sites in northern Arkansas.** Number of GPS collared adult white-tailed deer at three sites in a high CWD prevalence region of northern Arkansas from 2021 to 2024, separated by sex. Information from GPS collared deer was used to inform detection parameters used in abundance estimation.

| Site | Sex | 2021 | 2022 | 2023 | 2024 |
|------|-----|------|------|------|------|
| Erbie | Female | 1 | 4 | 4 | 4 |
| | Male | 3 | 4 | 1 | 2 |
| Gene Rush | Female | 23 | 22 | 22 | 15 |
| | Male | 8 | 9 | 9 | 4 |
| Tyler Bend | Female | 20 | 20 | 20 | 15 |
| | Male | 4 | 11 | 6 | 6 |

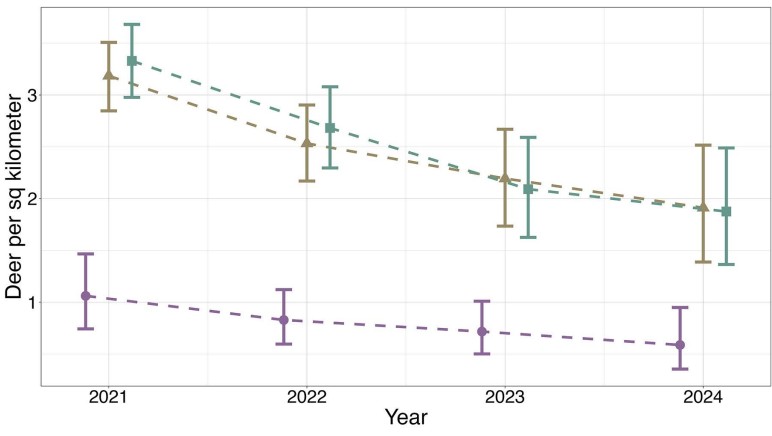

**Fig 2. Adult deer density per square kilometer.** Adult deer density per square kilometer for three sites in northern Arkansas for July 1 – July 14, 2021–2024. Error bars represent 95% credible intervals.

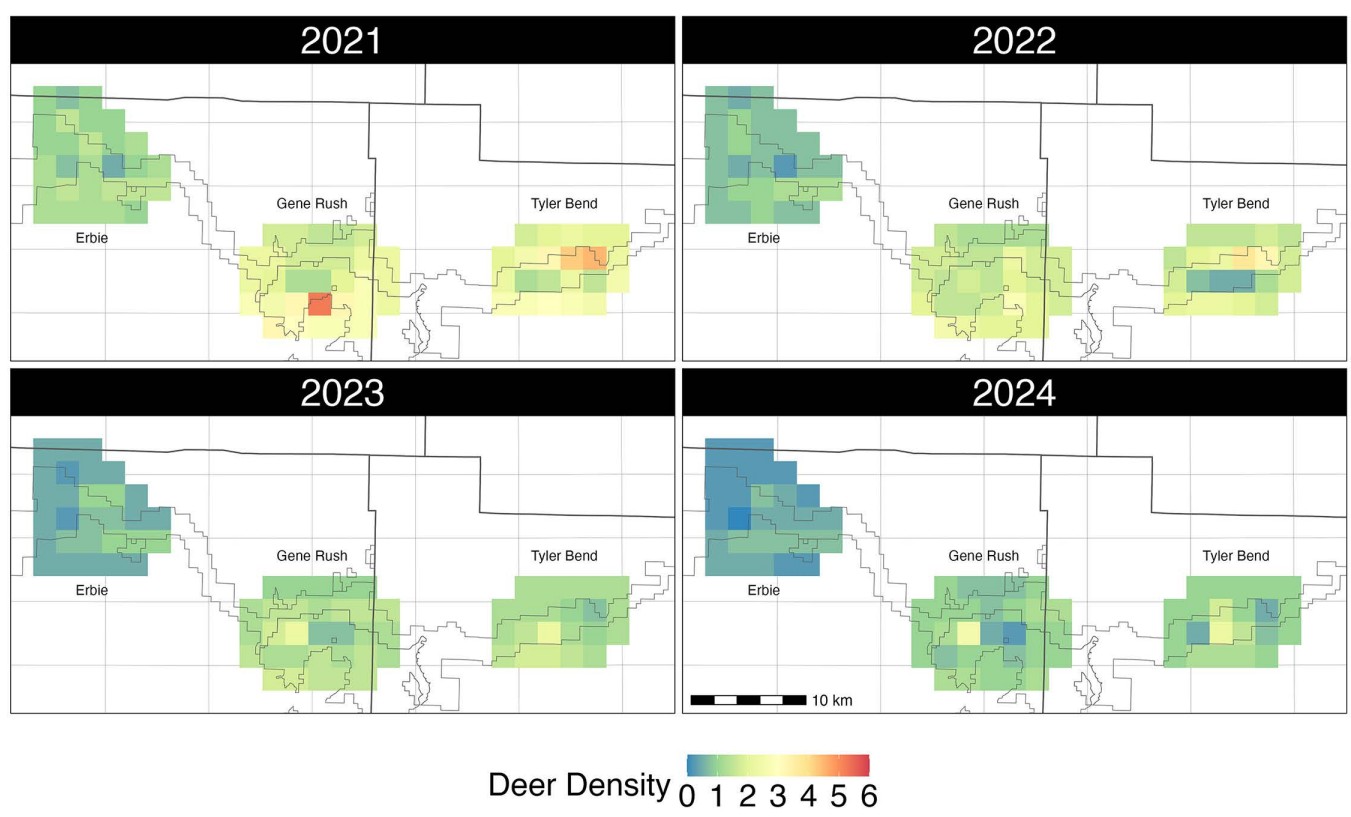

**Fig 3. Spatial deer density per square kilometer.** Adult deer density per square kilometer for three areas in Newton and Searcy Counties, Arkansas from July 1 – July 14, 2021–2024. Dark black lines depict county boundaries, with grey lines indicating state wildlife management areas.

**Table 3. Estimates of coefficients of expected density.** Estimates of coefficients of expected density for a spatial mark resight model of white-tailed deer in a high CWD prevalence region of northern Arkansas from July 1 – July 14, 2021–2024. Estimates describe the median and 95% credible interval, separated by sex and site where relevant. Astricts represent time trend credible intervals that do not contain 0.

| Parameter | Site | Stage | Estimate | 95% CI |
|---|---|---|---|---|
| $\alpha_0$ | | | 0.94 | 0.81–1.10 |
| $\alpha_1$ | | | −1.04 | −1.55 − −0.72 |
| $\alpha_2$ | | | 0.64 | 0.43–0.86 |
| $\alpha_3$ | Erbie | Female | −0.26 | −0.61–0.09 |
| | | Male | −0.25 | −0.57–0.07 |
| | Gene Rush | Female | −0.17 | −0.59–0.16 |
| | | Male | −0.39 | −0.80 − −0.7* |
| | Tyler Bend | Female | −0.12 | −0.25 − −0.01* |
| | | Male | −0.21 | −0.39 − −0.03* |

Gene Rush and Tyler Bend, average annual declines in abundance were 16% (95% CI: 6% − 24%) and 17% (95% CI: 8% − 26%) respectively.

We used the estimated male and female densities to calculate sex ratios at each site. At all sites, our median estimates suggested there were more females than males and male densities declined faster than female densities, but the differences between sexes were not significant (Fig 4). Yearly estimated sex ratios were imprecise and highly variable between years and estimates of the rate of decline at each site overlapped between sexes. The median estimates for all site and sex specific time trends were negative, but the 95% credible intervals for Erbie (both male and female) and females at Gene Rush included positive values. In 2021, median sex ratios were roughly equal across all sites (52%, 51% and 52% female deer at Erbie, Gene Rush and Tyler Bend respectively), compared to 59% female (56%, 59% and 60%) in 2024 (S1 Fig).

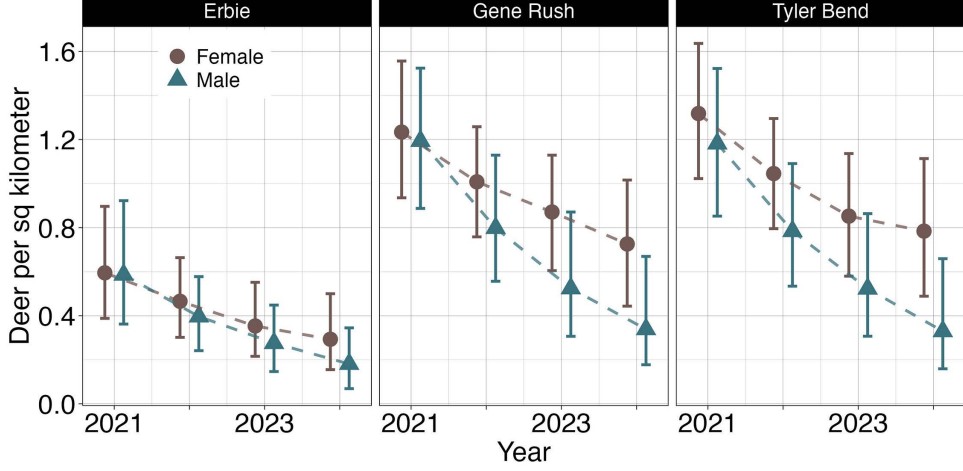

**Fig 4. Adult deer density per square kilometer by sex.** Adult deer density per square kilometer for three sites in northern Arkansas (Erbie, Gene Rush and Tyler Bend) for July 1 – July 14, 2021–2024, separated by sex. Error bars represent 95% credible intervals.

## Discussion

Deer densities at three sites in Arkansas's Tier 1 CWD zone declined over the four-year study period, corroborating previous work demonstrating declining cervid populations in areas with high CWD prevalence. Several studies on both mule deer and white-tailed deer have suggested that stable or increasing populations are unlikely when CWD prevalence exceeds 25–30% [5,6]. In Colorado, Miller et. al [4] documented an apparent 45% decline in mule deer abundance across a 20-year time frame when CWD prevalence was approximately 25%, despite the presence of suitable habitat, minimal hunting pressure, and survival rates in CWD-negative adults similar to those found elsewhere in the range. In a study on white-tailed deer in Wyoming, Edmunds et al. [5] documented a 10% annual decline for an area with 25% CWD prevalence. In light of this past research, the high apparent CWD prevalence and documented declines in our own study area suggest the deer population in northern Arkansas is not viable under current harvest levels.

To the best of our knowledge, no previous published studies have estimated deer densities in the Arkansas northern CWD management zone, with the exception of a long-term monitoring effort for an un-hunted deer population at Arkansas Post National Memorial in Benton County, Arkansas [36]. CWD was first detected in Benton County in 2018, but average sample prevalence was below 2% at the time of our study [28]. Mean deer densities at the Arkansas Post National Memorial declined from around 90 deer per square kilometer in 2020–55 deer per square kilometer in 2024, but the cause of these declines is unknown [36]. However, the high deer densities within the park are likely unrepresentative of the average deer population across the region, especially in comparison to public lands with high harvest pressure. Within the counties in our own study area, harvest records suggest hunter harvest for Newton County declined from 0.97 deer per square kilometer during the 2020–2021 season to 0.62 deer per square kilometer during the 2024–2025 harvest season, despite no changes in harvest limits or other management actions (36, S2 Fig). Additionally, harvest for Gene Rush WMA (located within Newton County) showed a steady decline from 1.53 deer per square kilometer during the 2020–2021 season to 0.85 deer per square kilometer during the 2024–2025 harvest season. In the same time period, harvest in Searcy County declined from 1.35 deer per square kilometer to 1.16 deer per square kilometer, but harvest in the intervening years was much more variable. Similarly, harvest across the broader deer management zone (Zone 2 as delineated by AGFC) also declined from 10,064 deer during the 2019–2020 deer season to 7,572 deer during the 2024–2025 season, with a low of 7,147 deer harvested during the 2021–2022 season [37]. While harvest trends do not always directly correspond to changes in abundance and may instead reflect changes in hunter effort [38], the observed trends suggest white-tailed deer abundance is likely decreasing both in the study area and across the broader CWD management zone.

While deer densities appeared to decline in response to high CWD prevalence in the region (S2 Fig), additional stressors may be contributing to deer population declines. In February 2021, weather stations in Newton County recorded a minimum temperature of −9°C, the coldest temperature on record since the 1930s [39]. Extreme temperature events can negatively impact deer abundance and productivity [40,41], but long-term population-level impacts are usually minimal unless extreme temperatures are sustained [42,43]. In addition to environmental factors, predation by coyotes (*Canis lupus*) or black bears (*Ursus americanus)* may have contributed to high fawn mortality and lower population growth rates [44,45], though no studies have documented a substantial increase in predator populations in this area during the study period. While coyote predation on healthy, adult white-tailed deer is relatively rare [46], reduced alertness in the later stages of CWD may elevate predator success and depress adult survival [4,47]. However, given the high sample prevalence in both Newton and Searcy Counties during our study (maximum reported prevalence 50% and 18% respectively), it is unlikely that weather or predation alone can explain the rapid declines in deer density. Rather, we hypothesize that declining deer densities reflect the negative population impacts of CWD, which may include higher mortality from predation and extreme weather events. Additional research is warranted to investigate the cause-specific mortality for deer in this population.

At the population level, CWD can be characterized by four epizootic stages – introduction and establishment (Stage 1), acceleration of incidence and prevalence (Stage 2), compounding prevalence (Stage 3), and enzootic equilibrium (Stage 4) [48]. In the early stages of CWD, when prevalence is low, population-level impacts may be minimal. However,

as prevalence rises, densities may decline substantially [5,48,49]. In our own study site, deer densities were lowest where presumed CWD prevalence was highest, but sex and site-specific time trends were not significantly different from 0. In contrast, deer declines were most severe at our most eastern site, Tyler Bend, with median deer densities declining more than 44% across the 4-year study and annual trend estimates that did not include 0. These differences may be explained both by the length of time CWD has been present at each of our study sites and possible density dependent transmission rates. Though not documented here, it is possible that the Erbie study site experienced similar declines prior to 2021 and is approaching a temporary equilibrium between disease prevalence and density-dependent transmission as prevalence reaches high levels [50]. Previous models of CWD suggest that after initial population declines, local CWD prevalence and transmission rates may be lower due to low population densities [2]. As the population reproduces and densities rise, prevalence increases and the cycle continues. Therefore, while our Erbie study site may show minimal declines in the short term, we predict future declines are possible once population densities rebound. Furthermore, we predict that population declines at both Gene Rush WMA and Tyler Bend will continue unless CWD prevalence is reduced or the population reaches lower densities.

Deer densities were negatively associated with percent pasture and grassland cover, consistent with previous research on deer habitat selection [25,51]. White-tailed deer often prefer locations within 200 m of forest edges, where forest cover provides opportunities for predator avoidance but higher quality forage is more abundant outside the shade limitations of closed-canopy forests [51,52]. Open fields, wildlife-openings and sparse canopy areas can provide high quality forage for deer, but are also attractive to both natural predators and hunters [53,54]. Avoidance of open areas is often highest during fawning season, decreasing as winter conditions limit available forage [55,56]. Thus, if we had focused on deer abundance earlier in the year, we may have detected a stronger relationship between landcover and deer densities.

The estimated declines in deer density in this region are some of the highest rates of declines ever reported for a cervid population with CWD. One plausible explanation is the high apparent prevalence (23%) in the region when CWD was first detected in Arkansas relative to initial prevalence estimates in other states (1–5% for Wisconsin, Colorado, and Pennsylvania, [57]). When CWD is detected early, targeted culling or adjustment of harvest limits may be effective at limiting population declines [58,59]. However, once prevalence reaches high levels, management options become more limited and substantial declines are more likely [5,6]. Our research underscores the importance of continued surveillance for CWD in areas without positive detections and the importance of rapid management responses to new detections.

With the spread of CWD to new regions of Arkansas, understanding how white-tailed deer populations are changing will be critically important for identifying effective management strategies. Accurate abundance estimates at the county, deer management zone, or state level will require auxiliary information from marked individuals. While the combination of telemetry data and camera data allow for accurate estimation of cervid populations [17], the resulting estimates often include large credible intervals that can mask subtle changes in population densities or sex ratios. If resources are available to uniquely identify male deer in camera photos across the state, these data could be analyzed with standard spatial capture-recapture models to estimate male densities. A subset of deer could also be physically marked with ear tags or livestock dye [60] prior to hunting season to allow for more precise estimates of hunting rates and abundance using integrated band-recovery models [61].

## Conclusion

Our findings add to a growing body of evidence that CWD can greatly alter deer populations. Given that previous research has demonstrated reductions in survival due to CWD [5], the high prevalence in our study area likely contributed to the observed population declines. However, it is possible that factors other than CWD contributed to the observed trends in deer densities. Future work in this region should document the direct effects of CWD on vital rates such as survival and fecundity to confirm the role of CWD in population declines. At present, few management tools are available to control CWD once established on the landscape. We therefore suggest that wildlife managers focus on slowing or preventing the

spread of CWD to uninfected populations, taking actions that aim to stabilize or lower CWD prevalence, and collect accurate data on deer abundance to monitor when harvest limits become unsustainable.

## Supporting information

**S1 Table. Estimates of encounter rate and spatial scale parameters.**
(DOCX)

**S1 Fig. Estimated sex ratio of adult white-tailed deer.**
(DOCX)

**S2 Fig. CWD sample prevalence and hunter harvest in Newton County, Searcy County and Arkansas's Zone 2 harvest area.**
(DOCX)

## Acknowledgments

We thank Arkansas Game and Fish Commission staff for administrative and logistical assistance. We thank the Warnell School of Forestry and Natural Resources at the University of Georgia for logistical support and numerous field techs for assistance with data collection. The views and opinions expressed herein are those of the authors and do not necessarily reflect the views or policies of the Arkansas Game and Fish Commission. Product references do not constitute endorsements.

## Author contributions

**Conceptualization:** Heather E. Gaya, Marcelo H. Jorge, Richard B. Chandler.

**Formal analysis:** Heather E. Gaya.

**Funding acquisition:** Mark G. Ruder, Gino J. D'Angelo, Richard B. Chandler, Michael J. Chamberlain.

**Investigation:** Marcelo H. Jorge, Lisa A. Jorge, Mark G. Ruder.

**Methodology:** Heather E. Gaya, Richard B. Chandler.

**Project administration:** Lisa A. Jorge, Gino J. D'Angelo, Michael J. Chamberlain.

**Resources:** Mark G. Ruder, Gino J. D'Angelo, Richard B. Chandler, Michael J. Chamberlain.

**Supervision:** Mark G. Ruder, Gino J. D'Angelo, Richard B. Chandler, Michael J. Chamberlain.

**Validation:** Heather E. Gaya.

**Writing – original draft:** Heather E. Gaya.

**Writing – review & editing:** Heather E. Gaya, Marcelo H. Jorge, Mark G. Ruder, Gino J. D'Angelo, Richard B. Chandler, Michael J. Chamberlain.

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
