## [Decision Letter · Decision Letter 0]

25 Sep 2025

We look forward to receiving your revised manuscript.

Kind regards,

Byron Caughey

Academic Editor

PLOS ONE

Journal Requirements:

“Funding for this project was provided by the United States Forest Service, CWD Alliance, Cabela Family Foundation, Boone and Crockett Club, the Wildlife Restoration Program through the U.S. Fish and Wildlife Service, and the Arkansas Game and Fish Commission (AR-W-F20AF00265).”

4. Please note that funding information should not appear in the Acknowledgments section or other areas of your manuscript. We will only publish funding information present in the Funding Statement section of the online submission form. Please remove any funding-related text from the manuscript. 

5. We note that you have indicated that there are restrictions to data sharing for this study. PLOS only allows data to be available upon request if there are legal or ethical restrictions on sharing data publicly. For more information on unacceptable data access restrictions, please see http://journals.plos.org/plosone/s/data-availability#loc-unacceptable-data-access-restrictions. 

6. Please note that your Data Availability Statement is currently missing the repository name. If your manuscript is accepted for publication, you will be asked to provide these details on a very short timeline. We therefore suggest that you provide this information now, though we will not hold up the peer review process if you are unable.

7. In this instance it seems there may be acceptable restrictions in place that prevent the public sharing of your minimal data. However, in line with our goal of ensuring long-term data availability to all interested researchers, PLOS’ Data Policy states that authors cannot be the sole named individuals responsible for ensuring data access (http://journals.plos.org/plosone/s/data-availability#loc-acceptable-data-sharing-methods).

8. We note that Figures 1 and 3 in your submission contain map images which may be copyrighted. All PLOS content is published under the Creative Commons Attribution License (CC BY 4.0), which means that the manuscript, images, and Supporting Information files will be freely available online, and any third party is permitted to access, download, copy, distribute, and use these materials in any way, even commercially, with proper attribution. For these reasons, we cannot publish previously copyrighted maps or satellite images created using proprietary data, such as Google software (Google Maps, Street View, and Earth). For more information, see our copyright guidelines: http://journals.plos.org/plosone/s/licenses-and-copyright.

1) You may seek permission from the original copyright holder of Figure 1 and 3 to publish the content specifically under the CC BY 4.0 license.  

2) If you are unable to obtain permission from the original copyright holder to publish these figures under the CC BY 4.0 license or if the copyright holder’s requirements are incompatible with the CC BY 4.0 license, please either i) remove the figure or ii) supply a replacement figure that complies with the CC BY 4.0 license. Please check copyright information on all replacement figures and update the figure caption with source information. If applicable, please specify in the figure caption text when a figure is similar but not identical to the original image and is therefore for illustrative purposes only.

**Additional Editor Comments:**

Two experts in the field have now reviewed your manuscript and, while they appreciated the issue being addressed and the extensive and valuable data that you have brought to bear the subject, they also each identified a number of major and minor shortcomings that should be considered and remedied to the extent possible. It is especially important for you to bolster the support for your conclusions with statistical analyses and mathematical modeling as they have suggested and to more clearly distinguish between well-supported conclusions and more speculative inferences/hypotheses. I invite you to submit in a revised version of the manuscript that addresses each of the points raised during the review process.

Reviewers' comments:

Reviewer's Responses to Questions

**Comments to the Author**

1. Is the manuscript technically sound, and do the data support the conclusions?

Reviewer #1: No

Reviewer #2: Partly

2. Has the statistical analysis been performed appropriately and rigorously?

Reviewer #1: No

Reviewer #2: No

3. Have the authors made all data underlying the findings in their manuscript fully available?

Reviewer #1: No

Reviewer #2: Yes

4. Is the manuscript presented in an intelligible fashion and written in standard English?

Reviewer #1: Yes

Reviewer #2: Yes

Reviewer #1: Review of PLOS ONE paper titled: White-tailed deer population declines in a high-prevalence chronic wasting disease region of Arkansas, USA by Gaya et al.

Review by Michael D Samuel

The goal of this paper is to determine if high prevalence of CWD reduces the population density at 3 different study sites within the CWD affected area of north west Arkansas. The study deployed camera traps at each study site and used spatial mark-recapture methods combined with auxiliary modeling to estimate annual deer density during 4 consecutive years (2021-2024). The authors present annual deer density at each of the 3 study sites. CWD prevalence data for the 3 study sites is only provided as summary information. Two of the study sites are reported to have higher prevalence than the third site. Although reported deer density estimates are highly variable the paper concludes that density was higher in the study site with lower CWD prevalence and deer density declined at all 3 study sites during the 4 years study.

Summary Review:

This study uses a novel approach for estimating deer density and evaluating changes in density associated with CWD prevalence/history. This is an important topic as it attempts to estimate the impact of a complex disease (CWD) on a socially and economically important wildlife species (white-tailed deer). I appreciate the extensive efforts, time, and resources required for this study. There is an extensive description of the SCR methodology used to estimate annual deer density at 3 different study sites over a 4-year period. This may be the first study to estimate the association between CWD and deer density. Although I am not a SCR expert I found this analysis to be confusing and not clearly described. I doubt that many readers would be able to replicate the analyses as described in the paper. I strongly recommend a diagram to show more clearly how the various model pieces integrate to estimate deer density and clear/consistent definition of model parameters (including units). Despite the extensive analysis it appears that density estimates are highly imprecise and are much lower than would be expected based on external deer abundance estimates and on harvest from these areas. Both these factors make me reluctant to accept the study conclusions of an association between deer density and CWD prevalence.

I found that the majority of the paper focused on methods used to estimate deer density while the spatial distribution/prevalence of CWD was largely ignored. The paper makes unsupported (assumed) spatial patterns of CWD distribution. In addition, CWD prevalence at the study sites is only vaguely reported using ranges of prevalence. The paper concludes, without scientific/statistical analysis, that deer density declined during the study. Given the lack of precision in estimated deer density I am skeptical such a conclusion is valid. Unless the authors can substantially improve the precision (and hopefully negative bias) of the density estimates I doubt a robust evaluation of density and CWD are possible. I recommend a much more rigorous evaluation of the landscape distribution and prevalence of CWD. The authors incorrectly conclude that an estimated stable population at one of the study sites results because CWD prevalence has reached stability. This conclusion is incorrect and needs to be reconsidered. Given the high prevalence typical of a stable equilibrium I don’t see why deer populations would remain stable? Finally, the authors should at least mention alternative hypotheses for population decline (harvest, predation, regional patterns) and absence of negative control (low/no CWD) in the study design. In particular, did increased harvest implemented as a management strategy cause population declines?

Specific review comments:

The authors hypothesize that deer densities would decline across the study period due to CWD and therefore deer densities would be negatively associated with CWD prevalence. However, alternative reasons for potential population decline are not considered.

Line 77-79: A CWD prevalence gradient was assumed based on a decline in the density of CWD detections in deer. While density of CWD detections could be a reliable surrogate for CWD this generally assumes similar spatial distributions of deer testing. Presentation of CWD prevalence data near the study sites would be a much better indicator of a prevalence gradient. I recommend the authors conduct a spatial analysis to quantify the landscape distribution of CWD for the study area to clarify and support their assumptions about spatial patterns.

Deer density was estimated using SCR methods and additional covariate modeling to account for distance from the initial CWD detection and the importance of pasture on deer density. I’m not an SCR expert so I can’t evaluate the technical components of this analysis. However, I had considerable difficulty integrating the different equations into a coherent framework. I strongly recommend providing a diagram to show how the various model parameters integrate and especially how they contribute to estimating density. At present I doubt that the current description would allow others to replicate the analysis. I was especially concerned because I was never able to clearly see how “z” values were estimated and/or how they were used to estimate the precision of “N”.

Line 127: why did you choose 2 x 2 km pixels? Looking at Figure 3 there seems to be considerable spatial variation in density within each study site. I assume this spatial variation contributes to variation in “N”? It seems to me larger pixels would tend to smooth spatial variation and improve the precision of “N”. Are their guidelines for making this decision?

Lines 120-132: why do you assume the mean easting value represents the first detection of CWD instead of using the actual location? It seems you are assuming this location is the origin of CWD in this region (significant parameter in results)? Given the high prevalence upon discovery it seems highly unlikely this location happens to be the origin and/or center of the outbreak. You need to show both points on Figure 1 and clearly describe the rational and interpretation for this variable.

Line 134: it seems like your pasture variable is basically trying to remove the area of non-deer habitat?

Line 148: not precisely clear what binary data for a day means: 0 for no deer, 1 for GE 1 deer (marked or unmarked)?

Lines 149-151: “k” needs to be defined. Is this a day? In general, I found some of your model parameters to be inconsistently/or not well defined. In addition, the units associated with the parameters need to be clearly defined.

Eq 3: How is “z” estimated – I found no equation for this variable? How was the precision of “N” estimated? Given the high variation of “N” in your figures you should make this clear.

Lines 165-167: provide the time range when does tend to isolate. Why does July 1- July 14 maximize doe detection? How does this affect male estimates? Will many males have antler development by this time so that males and females can be distinguished – yearlings problematic? Why not use larger time periods to help increase precision of your estimates? This is mostly confusing with no particular rational or evaluation.

Table 1: It isn’t at all clear how alpha1 and alpha2 directly impact deer density. See comments about better description and integration of model components.

Table 2: Given the low encounter rates (and low precision) it isn’t surprising that estimated density also lacks precision? Can the encounter/detection rates be improved? Why is it important to present these model parameters, but not others found in Table 1? Will these estimates be useful to readers – how?

Lines 191-193: only deer with GPS collars used to estimate detection rates? Clarify how unmarked deer contributed to density estimates as well.

Table 3: add how many detections of marked deer occurred.

Line 195: “CWD apparent prevalence in Newton County”

Lines 195-197: You should provide the annual prevalence data for these 2 counties. More importantly, you should provide the annual prevalence data surrounding your study sites. It seems like a spatial analysis of CWD detections is needed to understand the spatial pattern in detections/prevalence.

Lines 215-222: This short paragraph is the entire evaluation of the study hypothesis that CWD reduces deer density. It is crucial that statements made about deer density be rigorously justified using scientific/statistical analysis. Given the poor precision for density estimates inf Figs 2 & 3 simple statements based on median estimated density could be misleading. The study found a significant relationship for the easting parameter, but it isn’t clearly stated what this means – study sites further east had higher density? Statements in this paragraph about changes in deer density or density changes must be supported by statistical analysis that consider both median density estimates and their associated precision. Based on the high variation in estimated density I’m very skeptical that reported results are valid. Although the data is suggestive for the Gene Rush study site. I recommend that density changes over time be evaluated using trend analysis (regression) such that annual changes are statistically evaluated and quantified. Finally, the estimated deer densities in this paper are considerably lower than average deer densities in Arkansas (about 6 deer/km). Is this related to the study design? This needs some discussion about potential bias.

Lines 231-235: it isn’t clear how this paragraph contributes to the paper. Given the lack of precision in estimated density it not surprising that density of males or females did not significantly decline. Please provide estimated precision for % female abundance. It seems surprising that female density would be lower than male density – as most deer populations are the opposite due to male harvest?

Line 242: this statement needs to be supported by a statistical analysis!

Line 248: minimal hunting but substantial predation

Line 249: suggest saying 10% annual decline instead of .9 growth rate

Lines 253-269: this paragraph is important because it raises alternative hypotheses about why deer density may also change at the study sited. First, study lacks a negative control (low or no CWD) so trends at Post National are important. This area report about 10% annual decline in deer density over the similar time period. Could regional deer density be declining due to causes other than CWD (predation, harvest, etc.)? Also why are deer densities so much higher than your study sites? Harvest records in this paragraph are also useful because they demonstrate changes in harvest rates that could also influence annual deer density. Are some of the patterns you see affected by deer harvest? How can this factor be considered in your evaluation? Finally, the fact that annual hunter harvest exceeds your density estimates makes me question whether your estimates are valid. You need explain this discrepancy.

Lines 275-281: you have likely misinterpreted Samuel (2023). There is no connection between stable equilibrium and deer density. Density is likely to decline with the high prevalence at Stage 4; however, this decline is independent of transmission dynamics and related to high prevalence. So, reaching Stage 4 does not mean that populations will become stable. Stable levels of infection are based on the population proportions that are infected, survival rates, birth rates of uninfected fawns, and transmission rates and not on deer density. Stage 4 can occur at different population densities because transmission is not density dependent. In summary, reaching stable CWD prevalence is not a sufficient reason for population stability.

Lines 284-292: the paper provides a discussion and interpretation of the pasture parameter (alpha1). However, there is no similar discussion of the distance parameter (alpha 2) which seems much more relevant to the CWD-deer density hypothesis. How should alpha2 be interpreted relative to deer density and CWD distribution?

Figure 1: please add study site and county names to this figure. You should also add the first CWD detection and the mean easting used to calculate “easting” values. Providing an assumed gradient for CWD landscape distribution seems rather arbitrary. As recommended above a spatial analysis of CWD detection/prevalence would be preferred.

Figure 2: use “Estimated Deer…..”

Figure 3: please add study site and county names to the figure. Label should be “Estimated Deer Density”

Figure 4: use “Estimated Deer ….”

Reviewer #2: Dear Authors,

Your paper "White-tailed deer population declines in a high prevalence chronic wasting disease region of

Arkansas, USA" compiles an enormous amount of data collected from camera traps and GPS collaring to estimate deer density in three regions of Arkansas that differed in CWD prevalence. The premise of this study is to determine the effects of CWD on deer populations depending on disease prevalence. The data could be very helpful for deer management decisions in CWD endemic regions.

Comments:

Major comments: Figure 2 (and related text and conclusions), the error bars for the deer density data are extremely large and often overlapping in these data sets. I could not find any statistics on these comparisons, but aside from the starting density and ending density in the Gene Rush regions I doubt any of these values are statistically different. Did you apply any statistical tests to the data beyond error bars? Efforts to interpret this data should be expanded, even if the statistics suggest the population densities are not changing in two of the areas that should be made clear.

Conclusions in lines 250-252 and 30-31 would benefit from some mathematical modelling or more detailed numbers to back up the claim that populations cannot be sustained under the current regulations. This may in fact be true, but it is impossible to assess with the limited data currently shown. It is not clear how much hunter harvest is actually occurring in these areas compared to other sources of depredation and disease. If hunter harvest is a very small percentage of the deer removed, then perhaps the population won't be as affected? For example, if CWD is killing 13% of the population but hunter harvest is only 5% of the population then long term population effects will be much more dependent on CWD. On the contrary, if CWD is killing 13% yearly and hunters harvest is 35% yearly then I agree that the populations cannot sustain these additive pressures. However, without more specific numbers for the different sources of deer deaths (or % of the deer population) and the expected recruitment rate of fawns in these areas it is very hard to understand how these conclusions were drawn.

Minor comments:

Lines 26-28 and 220-222: Providing the final deer density values has little to no meaning to the reader in these locations as there are no starting values provided in the adjacent text to provide context.

Figure 1: Label the three study regions by the names used elsewhere in the manuscript.

Can the authors comment on whether any major environmental effects or disease outbreaks occurred in the study regions during the study period, such as drought, wildfire, hurricanes, excessive rainfall during fawning season, EHD etc. that may have played a role? This seems especially important since there was discussion about an a unit in Benton County that experienced similar deer declines despite having minimal CWD (lines 257-259).

Lines 264-267. The data on decreasing harvest numbers is helpful for discussion but would be stronger if you also included the hunter effort during these same years... If hunters declined you would anticipate harvest decline, and vice versa. Was hunter effort consistent? Was expected predation by non-humans fairly consistent?

**Do you want your identity to be public for this peer review?** For information about this choice, including consent withdrawal, please see our Privacy Policy

Reviewer #1: **Yes: ** Michael D Samuel

Reviewer #2: No

---

## [Author Response · Author response to Decision Letter 1]

12 Nov 2025

We have attached a document responding to all reviewer comments. The editor did not make any specific comments about the manuscript.

---

## [Decision Letter · Decision Letter 1]

21 Nov 2025

Dear Dr. Gaya,

Thank you for your revision, which both reviewers agree has improved your manuscript. However, one still has a number of concerns of a more minor nature that I would invite you to address prior to our further consideration.

We look forward to receiving your revised manuscript.

Kind regards,

Byron Caughey

Academic Editor

PLOS ONE

**Journal Requirements:**

Reviewers' comments:

Reviewer's Responses to Questions

**Comments to the Author**

Reviewer #1: (No Response)

Reviewer #2: All comments have been addressed

2. Is the manuscript technically sound, and do the data support the conclusions?

Reviewer #1: Yes

Reviewer #2: Yes

3. Has the statistical analysis been performed appropriately and rigorously?

Reviewer #1: Yes

Reviewer #2: Yes

4. Have the authors made all data underlying the findings in their manuscript fully available?

Reviewer #1: Yes

Reviewer #2: Yes

5. Is the manuscript presented in an intelligible fashion and written in standard English?

Reviewer #1: Yes

Reviewer #2: Yes

Reviewer #1: Review of revised PLOS ONE paper titled: White-tailed deer population declines in a high-prevalence chronic wasting disease region of Arkansas, USA by Gaya et al.

Summary Review:

I believe the paper as improved substantially from the first version. Improvements include better description of SCR methods, trend evaluation for deer density estimates, consideration of other potential factors affecting deer populations, additional data on CWD prevalence and harvest data.

Overall, I believe some additional relatively minor improvements could be made to further improve the paper. I believe there is need for additional clarification of the SCR equations, in particular the time parameter (t). A sex and site-specific discussion of population trends would seem important with more specifics on estimation of the average annual trend and estimated CI – readers should be able to replicate your results. I have remaining concerns about the estimated sex-specific abundance results and trends relative to CWD prevalence. While adding a discussion of alternative population reduction mechanisms was helpful these could be reduced as they tend to distract from the larger results. The mention of density dependent transmission at the Erbie site is interesting and would benefit from further discussion/evaluation. This is an important aspect of CWD epizootiology that deserves more discussion. In general, I believe the paper would provide a stronger scientific contribution if it can generalize its findings relative to CWD management in both Arkansas and other CWD affected areas: harvest recommendations, CWD epizootiology, deer population decline from CWD, etc.

Specific review comments:

Line 53: harvest limits

Line 123: I believe you mean a starting date of July 2021?

Lines 122-124: consider moving this sentence to near 135-137?

Lines 128-133: I still don’t like this somewhat arbitrary assumption about the distribution. Based on the Figure S1 it seems you could at least add that Newton County has much higher prevalence than Searcy – generally supporting your decision.

Line 143 and later: the parameter t is not clearly defined. Later in the paper it seems to be defined as a 24 hr period between July 1 and July 14. Can you help the reader by giving better definition of t so it isn’t confused with annual change? Also, later in the paper you need to clarify if t is year-specific? If so it seems your equations with t would be year specific? For example, in EQ 1, if t is not year specific then how is it defined? Overall, I’m puzzled what specific time interval t represents. Also see comments below.

Line 152: I believe you need to show a specific equation here – is this actually EQ 3? This is the crucial equation for estimating annual changes in density. Because this equation has a trend parameter it is not simply similar to EQ 1 and should be shown. Also, for time trends in Table 1 and later I suggest you use annual trends to distinguish from parameter t in your equations.

Table 1: The “a2” parameter is described as being relative to the first CWD detection, but your rebuttal comments indicated the first detection was west of Erbie – please clarify. It isn’t obvious why Erbie has a 0 prior on 2 parameters. Can you add a footnote to clarify this. For ‘a3” do you mean annual trend? Again, my confusion on t in your equations.

Line 165: please add the location for the easting point to Fig 1 so it is specifically defined. Without this location your results relative to easting can’t be replicated by others.

Line 169: I’m perplexed by this equation. Why is the “a3” part multiplied by (t-1)??? Do you mean evaluated at (t-1)? This part of EQ 3 is confusing and needs both clarification and explanation. If ‘a3’ represents a time trend (slope) how do you estimate first year (intercept? t=0?) sex and site-specific density? Also, I’m curious why you didn’t evaluate site-specific pasture parameters?

Line 174: is this the definition of t, a 24-hour period between July 1 and July 14? So, t is year specific and goes from 1-15 each year? Alternatively, t goes sequentially across 4 years 1-60? Or maybe t counts the elapse days since July 1, 2021? Again, I’m clearly missing something???

Lines 200-204: Please cite a reference for this information here and in Fig S1.

Table 2: in my view these are rather specialized parameters and could be reported in an appendix for interested parties.

Line 218: density was high at Grand River and Tyler Bend. This difference from Erbie deserves some discussion. Is it because of high CWD prevalence along your gradient? Does this provide further evidence of CWD impacts?

Lines 222-225: I suggest you reverse these two ideas. First describe site and sex patterns both significant and not, as well as quantitative declines (with CIs). Then present the average decline (with CI) and describe how that average was calculated. Also, how the average (with CI) for each site was calculated for Fig 2. If that description is too technical it may need to be in the methods section?

Table 4: Sex rather than Stage?

Lines 252-254: Given that 50% of your site/sex specific estimates seems to be equivocal (overlap 0) I think you should temper your statements here. In addition, this seems like a good place to discuss your more specific results and interpretation for site and sex differences. If your interpretation is that all sites are declining then some rational for that conclusion vs the typical 95% criteria should be concluded.

Lines 254-263: seems like this comparison should be a new paragraph. Note that lions may have played an important part in the Colorado deer decline. Also, the declines you estimated for your study are considerably larger than the other studies you mention (or that I’m aware of). For example, a deer population would be reduced by 50% every 2 years using your results. No one has documented such a rapid decline and this deserves some discussion. Is part of your decline potentially due to harvest and/or factors?

Lines 261-286: I don’t disagree that the reductions you report suggest changes in harvest are needed. I wonder if you can make a stronger case for the hunting impact on decline and/or how much harvest needs to be reduced? Reduce male harvest, female harvest, both? Given the harvest data in Fig S1 can you approximate what proportion of the decline might be caused by harvest? Harvest data is covered in lines 272-280 but can you better integrate these with your results. Although CWD and harvest are surely partly compensatory you might be able to suggest how much harvest is viable?

Lines 264-272: it seems to me that the 2 main messages here are that 1) deer density is much lower in the CWD area/your study area than many other parts of the state and 2) one other area has shown an unknown population decline not related to CWD. The later introduces the notion of other factors which follows. The first message brings up the potential that CWD may have different epizootiology and population implications based on deer density (see later comments).

Lines 287-303: I’m glad you added this type of information, but I suggest it be condensed substantially. These are other possibilities that might contribute to population decline, but likely much less important than high CWD and/or harvest.

Line 308: as prevalence reaches high levels

Line 309: our Erbie study site

Line 312: annual trend

Line 313-318: I don’t believe that reference 50 shows DD transmission. Instead, I suggest you review Storm et al (2013; Ecosphere). This paper indicates that CWD is generally FD at higher deer densities but DD at low densities – as far as I know this is the only study with data that shows potential for DD transmission? I think your study areas would be characterized as low density relative to Wisconsin. Would this provide a justification for DD transmission in your areas, but perhaps other patterns in higher density areas – similar to Wisconsin (Samuel 2023)? In addition, DD transmission doesn’t produce a sable equilibrium as you suggest. It typically means high prevalence reduces population levels, which reduces transmission rate and prevalence, then populations increase and the cycle starts again. See Almberg’s 2011 modeling paper. I think your predictions should try to account for these predicted D transmission/population patterns. If you can pull all this together it could make a very useful contribution of how CWD works in low density populations.

Fig S1: please provide citation(s) for the data in this figure.

Fig 1: please add the mean easting point to this figure. In my opinion your gradient is artificial and unnecessary. I think the county level prevalence data in Fig S1 is sufficient to show western areas have higher CWD. The easting point location is needed so others can replicate your analysis.

Fig 2: does not require colors and could be done in BW with different markers for each study site.

Fig 3: these detailed data are ancillary to the study and could be moved to supplementary material

Fig 4: can be done in BW. I have remaining concerns about the data in these figures. First, most hunted deer populations have considerably higher female deer abundance than males due to selective harvest. For example, males are typically about 1/3 of the population. Do you have data that would suggest equal abundance for males and females is typical? Second, your first year estimates show equal density of males and females, followed by males appearing to decline faster than females. This suggests males are more “vulnerable” to CWD than females, which is quite consistent with the literature. However, given the apparently high CWD prevalence in Newton County, why hasn’t CWD already reduced male abundance compared to females in 2 of your study sites? Both of these results reduce my comfort level in your results. I think you should address these concerns in the discussion.

Reviewer #2: Thank you for your efforts to improve your manuscript. The results and figures are much more clear now, providing a cohesive and easily interpreted message.

**Do you want your identity to be public for this peer review?** For information about this choice, including consent withdrawal, please see our Privacy Policy

Reviewer #1: No

Reviewer #2: No

---

## [Author Response · Author response to Decision Letter 2]

4 Dec 2025

We have uploaded our response to reviewer comments during the attach files section of manuscript submission.

---

## [Decision Letter · Decision Letter 2]

16 Dec 2025

White-tailed deer population declines in a high-prevalence chronic wasting disease region of Arkansas, USA

PONE-D-25-49401R2

Dear Dr. Gaya,

We’re pleased to inform you that your manuscript has been judged scientifically suitable for publication and will be formally accepted for publication once it meets all outstanding technical requirements.

Kind regards,

Byron Caughey

Academic Editor

PLOS One

Additional Editor Comments (optional):

Reviewers' comments:

Reviewer's Responses to Questions

**Comments to the Author**

Reviewer #1: All comments have been addressed

2. Is the manuscript technically sound, and do the data support the conclusions?

Reviewer #1: Yes

3. Has the statistical analysis been performed appropriately and rigorously?

Reviewer #1: Yes

4. Have the authors made all data underlying the findings in their manuscript fully available?

Reviewer #1: Yes

5. Is the manuscript presented in an intelligible fashion and written in standard English?

Reviewer #1: Yes

Reviewer #1: (No Response)

**Do you want your identity to be public for this peer review?** For information about this choice, including consent withdrawal, please see our Privacy Policy

Reviewer #1: No

---

## [Editor Report · Acceptance letter]

PONE-D-25-49401R2

PLOS One

Dear Dr. Gaya,

I'm pleased to inform you that your manuscript has been deemed suitable for publication in PLOS One. Congratulations! Your manuscript is now being handed over to our production team.

Kind regards,

on behalf of

Dr. Byron Caughey

Academic Editor

PLOS One